# Cancer cells with high-metastatic potential promote a glycolytic shift in activated fibroblasts

Akiko Kogure[1☯], Yutaka Naito[2☯], Yusuke Yamamoto[3], Masakazu Yashiro[4], Tohru Kiyono[5], Kazuyoshi Yanagihara[6], Kosei Hirakawa[4], Takahiro Ochiya[1]*

1 Department of Molecular and Cellular Medicine, Institute of Medical Science, Tokyo Medical University, Tokyo, Japan, 2 Tumor Cell Biology Laboratory, The Francis Crick Institute, France, United Kingdom, 3 Division of Cellular Signaling, National Cancer Center Research Institute, Tokyo, Japan, 4 Molecular Oncology and Therapeutics, Osaka City University Graduate School of Medicine, Osaka, Japan, 5 Division of Carcinogenesis & Cancer Prevention, National Cancer Center Research Institute, Tokyo, Japan, 6 Division of Biomarker Discovery, Exploratory Oncology Research & Clinical Trial Center, National Cancer Center, Tokyo, Japan

☯ These authors contributed equally to this work.
* tochiya@tokyo-med.ac.jp

**Data Availability Statement:** Raw and normalized microarray data are available in the Gene Expression Omnibus database (accession numbers GSE116167 and GSE116176).

## Abstract

Cancer-associated fibroblasts (CAFs) are activated fibroblasts and are the major stromal component in various types of malignancies. CAFs often undergo metabolic reprogramming to create an appropriate microenvironment for cancer progression. However, it remains unclear whether the metastatic properties of cancer cells affect aerobic glycolysis in stromal cells. Here, we show that gastric cancer (GC) cells with high metastatic potential strongly promote the metabolic switch from oxidative phosphorylation to aerobic glycolysis in fibroblasts. Transcriptome analysis showed that the expression of glycolysis-related genes, such as LDHA and ENO2, significantly changed in fibroblasts when they were cocultured with cancer cells with high metastatic potential compared to fibroblasts incubated with cancer cells with low metastatic potential. Glucose uptake, lactate production and oxygen consumption in fibroblasts were changed by coculture with GC cells with high metastatic potential. Thus, metabolic reprogramming in CAFs may reflect the metastatic properties of GC cells.

## Introduction

Cancer-associated fibroblasts (CAFs) are the major stromal components in various types of malignancies [1–3]. They are a heterogeneous population and originate from several stromal cell types, including resident fibroblasts, endothelial cells, pericytes, and bone marrow-derived cells [1,4]. CAFs play a pivotal role in cancer development and progression by enhancing cancer cell invasion, remodeling the extracellular matrix and mediating the inflammatory environment [1–3]. Several reports have suggested that stromal components, including CAFs, are

**Funding:** This work was supported by JSPS KAKENHI Grant Number 15K21646, 17H04991 and 17K19625, and JSPS Fellows 15J10473 and in part by a Grant-in-Aid for the Project for Cancer Research And Therapeutic Evolution (P-CREATE, Grant Number 16cm0106402h0001) and the Project focused on developing key technology for discovering and manufacturing drugs for next-generation treatment and diagnosis (Grant Number 16ae0101011h0003) from the Japan Agency for Medical Research and Developmentfrom, AMED.

**Competing interests:** No authors have competing interests.

potential therapeutic targets for cancer treatment and could be used to improve cancer diagnosis [5–7].

Growing evidence suggests that CAFs undergo aerobic glycolysis and produce various types of metabolites to support tumor growth [8]. Lactate is shuttled from CAFs via MCT4, and that mechanism is utilized by cancer cells for promoting the Krebs cycle as well as anabolic metabolism and cell proliferation [8]. It has also been reported that reprogrammed CAFs in the metastatic niche by tumor-derived extracellular vesicles (EVs) create a metabolic environment that supports tumor metastasis [9–11]. Targeting strategies of tumor stroma might diminish the nutrient balance within the tumor, leading to novel effective therapies for cancer treatment. However, several studies demonstrated that CAFs have many various subtypes with distinct functions and phenotypes during cancer progression [12–15]. These CAF phenotypes differ according to the cancer subtype and aggressiveness [13,15]. Although it is possible that the malignant properties of cancer cells exert a different effect on glucose metabolism in fibroblasts within the tumor microenvironment, it remains unclear whether the metastatic potential of cancer cells can be used to metabolically reprogram surrounding stromal cells.

We previously reported that highly metastatic gastric cancer (GC) cells more strongly affected the fibroblast phenotypes than did GC cells with low metastatic potential [15]. In this model, highly metastatic GC cells induced glycolysis-related gene expression in fibroblasts, but GC cells with low metastatic potential did not, suggesting that highly metastatic cancer cells potentially reprogram the metabolic status of surrounding fibroblasts. Here, with a metastatic model of diffuse-type gastric cancer (DGC), we investigated how the difference in glucose metabolism appeared by comparing DGC cells with high and low metastatic potential. Our study revealed that cancer cells with high metastatic capacity can strongly induce metabolic reprogramming. These results might provide novel information regarding the formation of a metabolic microenvironment for tumor metastasis.

## Materials and methods

### Cell cultures

HSC-44PE and 44As3 cell lines were used as described previously [16]. Two human normal gastric fibroblasts (NFs) were immortalized by infection with retroviruses expressing mutant Cdk4, cyclin D1 and human telomerase reverse transcriptase, as previously described; the resultant lines were named iNF-58 and iNF-60 [15]. Briefly, GC cell lines were cultured in RPMI-1640 (Thermo Fisher Scientific, Rockford, IL) supplemented with 10% heat-inactivated fetal bovine serum (FBS, Gibco, Thermo Fisher Scientific) and 1% Antibiotic-Antimycotic (Thermo Fisher Scientific) at 37˚C in 5% $CO_2$. iNF-58 and iNF-60 cells were cultured in Dulbecco's modified Eagle medium (DMEM; Thermo Fisher Scientific) with 10% FBS and 1% Antibiotic-Antimycotic (Thermo Fisher Scientific) and 0.5 mM sodium pyruvate (Sigma-Aldrich, MO, USA). For transwell cocultures, $3x10^4$ HSC-44PE or 44As3 cells were seeded into the top of a transwell membrane (0.4 μm pore size, Corning Life Science, Tewksbury, MA, USA) with iNF-58 or iNF-60 ($3x10^4$) growing in the lower compartment of a 6-well plate in DMEM (Thermo Fisher Scientific) with 10% FBS and 1% Antibiotic-Antimycotic (Thermo Fisher Scientific). Cell lines were tested for mycoplasma by PCR (e-MycoTM VALiD Mycoplasma PCR Detection Kit, iNtRON, Seoul, Korea).

### Microarray

mRNA and miRNA microarrays were performed as described previously [15]. Raw and normalized microarray data are available in the Gene Expression Omnibus database (accession numbers GSE116167 and GSE116176).

## RNA extraction and quantitative RT-PCR (qRT-PCR)

Total RNA was extracted from cancer-derived EVs (10 μg) and cultured cells using QIAzol and a miRNeasy Mini Kit (Qiagen, Hilden, Germany) according to the manufacturer's protocols. For mRNA expression by qRT-PCR analysis, complementary DNA (cDNA) was generated from total RNA using a High Capacity cDNA Reverse Transcription Kit (Thermo Fisher Scientific). Real-time PCR analysis of the cDNA was subsequently performed in triplicate using Platinum SYBR Green qPCR SuperMix UDG (Thermo Fisher Scientific). The data were collected and analyzed using a StepOne Real-Time PCR System and StepOne Software v2.3 (Thermo Fisher Scientific). All mRNA quantification data from cultured cells were normalized to the expression of β-actin (ACTB). The primers used in this study are as follows:

ACTB Forward: 5′-TCACCGAGCGCGGCT-3′, ACTB Reverse: 5′- TAATGTCACG-CACGATTTCCC-3′, ENO1 Forward: 5′-TGTGAGCCTCGTGTCATCTC-3′, ENO1 Reverse: 5′-CATGGGTCACTGAGGCTTTT-3′, ENO2 Forward: 5′-CATTGAGGACCCATTTGACC-3′, ENO2 Reverse: 5′-CAGTTGCAGGCCTTTTCTTC-3′, LDHA Forward: 5′-TAGTTCTGCCACCTCTGACG-3′, LDHA Reverse: 5′-AAACATCCACCTGGCTCAAG-3′, PDK1 Forward: 5′-GGATTGCCCATATCACGTCT-3′, PDK1 Reverse: 5′-ACTG-CATCTGTCCCGTAACC-3′, PDK3 Forward: 5′-GGTGGTTTATGTGCCCTCAC-3′, and PDK3 Reverse: 5′-AGCAGGGTAGCCCTCTTTTC-3′.

## Analysis of lactate production, glucose consumption and pH

Lactate production and glucose consumption levels in the medium were measured with a lactate assay kit (BioVision Technologies, Exton, PA, USA) and a glucose assay kit (Abcam, MA, USA) according to the manufacturer's instructions. Briefly, upper chambers containing cancer cells were removed from the wells, and were washed with PBS. Then, the flesh DMEM medium with 1g/L glucose without FBS was added to lower chamber containing fibroblasts. Three hours later, the supernatants were collected, and used for lactate production and glucose consumption assay. For checking medium pH, upper chambers containing cancer cells were removed from the wells, then we collected cultured medium from the lower well. The pH of the medium was measured using pH meters.

## Mitochondrial Oxygen Consumption Rate (OCR) determination

The OCR was determined using the commercial MitoXpress Xtra Oxygen Consumption Assay (Agilent) according to the manufacturer's instructions. Fluorescence intensity (excitation light: 380 nm; measured light: 645 nm) was detected with the Synergy H1 microplate reader (BioTek, Winooski, USA) at 2-min intervals for 120 min, and subsequent conversion to lifetime values was performed using Gen5 software (BioTek).

## Seahorse metabolism assay

OCR and extracellular acidification rates (ECAR) for iNF58 cells were determined using a Seahorse Extracellular Flux (XF96) analyzer (Seahorse Bioscience, North Billerica, MA, USA). Briefly, $1x10^4$ mono-cultured or cocultured iNF58 cells were seeded on XF96-well cell culture plates for 24 hours and incubated overnight at 37˚C in a 5% $CO_2$ humidified atmosphere. The next day, the cells were washed in prewarmed XF assay media. Cells were then maintained in 180 μL/well of XF assay media at 37˚C in an incubator without $CO_2$ for 1 hour. Subsequently, the OCR and ECAR were measured according to the manufacturer's instructions for the Seahorse XF96 Extracellular Flux Analyzer.

## Cell proliferation assay

For transwell cocultures, $1\times10^4$ HSC-44PE or 44As3 cells were seeded into the top of a transwell membrane with iNF-58 ($1\times10^4$ cells) growing in the lower compartment of a 24-well plate in DMEM with 10% FBS and 1% Antibiotic-Antimycotic. The cells were incubated for 1, 2 and 4 days, and then a CellTiter-GLo assay (Promega, WI, USA) was performed to assess cell proliferation.

## Immunoblot and densitometric analysis

For immunoblot analysis, whole-cell lysates were prepared with Mammalian Protein Extract Reagent (M-PER; Thermo Fisher Scientific). The whole-cell lysates (20 μg) were solubilized in Laemmli sample buffer by boiling, and then they were loaded onto a Mini-PROTEAN TGX Gel (4–15%, Bio-Rad, CA, USA) and separated by electrophoresis (100 v, 30 mA). The proteins were transferred to a polyvinylidene difluoride membrane (Millipore, Billerica, MA, USA). After blocking in Blocking One reagent (Nacalai Tesque, Kyoto, Japan), the membranes were incubated for 1 h at room temperature with the following primary antibodies: anti-LDHA (3582S, dilution 1:1000, Cell Signaling Technology (CST), Danvers/Massachusetts), anti-PDK1 (3062S, dilution 1:1000, CST), anti-PDK3 (ab182574, dilution 1:1000, Abcam), anti-actin (MAB1501, dilution 1:1,000, Millipore), anti-ENO1 (3810S, dilution 1:1,000,CST), anti-ENO2 (8171S, dilution 1:1,000, CST), anti-HIF-1α (#610959, dilution 1:500, BD Bioscience), and anti-HIF-2α (#NB100-122, dilution 1:1,000, Novus Biologicals, Littleton, CO, USA). Secondary antibodies (horseradish peroxidase-linked anti-mouse IgG, NA931 or horseradish peroxidase-linked anti-rabbit IgG, NA934, GE Healthcare, Milwaukee, WI) were used at dilutions of 1:5,000. Membranes were then exposed to ImmunoStar LD (Wako, Osaka, Japan), and then visualized with a ChemiDoc imager (Bio-Rad). The band intensities of ENO2, LDHA, PDK3, and β-actin were quantified with densitometry using Image J software (National Institutes of Health, Bethesda, USA).

## Statistical analysis

Values are represented as the mean ± s.d. for technical replicates. Statistical analyses between two groups were performed using Student's $t$-tests. One-way analysis of variance (ANOVA) was used to determine significant differences among three groups, followed by Tukey honestly significant difference (HSD) post hoc comparisons. A $p$-value less than 0.05 was considered statistically significant.

# Results

## GC cells with high metastatic potential strongly induced glycolysis-related gene expression in surrounding fibroblasts

To investigate whether cancer cells with high metastatic potential have the potent capacity to drive metabolic reprogramming of fibroblasts, we used transcriptome data we previously collected [15]. We used two types of DGC cell lines, HSC-44PE, a parental cell line with low metastatic potential and 44As3, a cell line with high metastatic potential [16], which was used as a metastatic model. We cultured immortalized human stomach fibroblast lines (iNF-58 and iNF-60) with these cancer cell lines using a transwell culture system (Fig 1A). Then, we performed whole transcriptome profiling of fibroblasts in mono-culture and coculture [15]. Gene set enrichment analysis (GSEA) revealed significant enrichment of genes with a role in "glycolysis" functions in fibroblasts cocultured with 44As3 cells compared to fibroblasts that were in mono-culture and those that were cocultured with HSC-44PE cells (Fig 1B, upper left and middle). The "glycolysis" pathways also enriched in fibroblasts cocultured with HSC-44PE compared

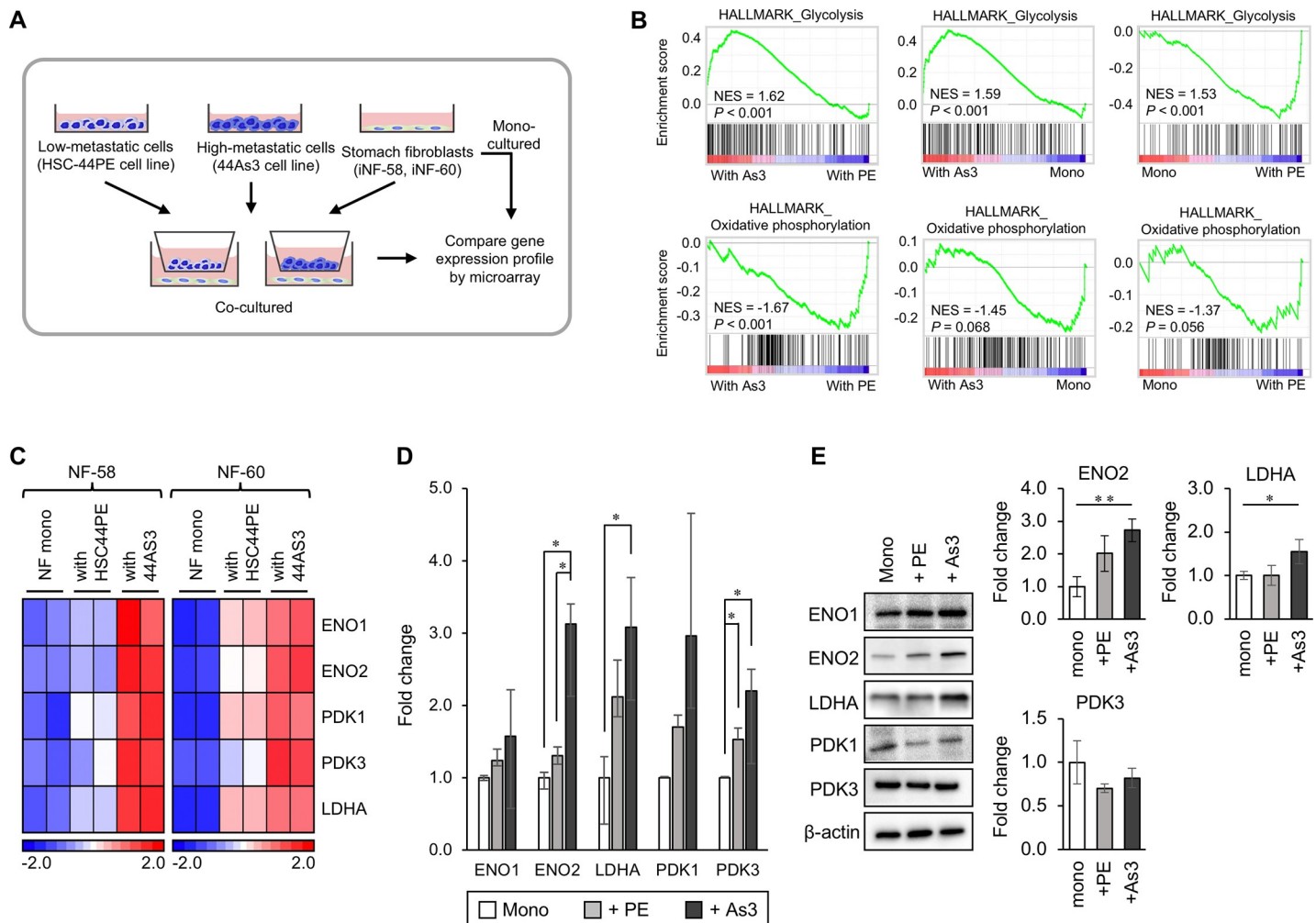

**Fig 1. Gastric cancer cells with high metastatic potential increased the expression level of glycolysis-related genes in the cocultured fibroblasts. (A)** Schematic protocol of coculture and gene expression analysis. (**B**) GSEA of the fibroblasts cocultured with 44As3 versus those cocultured with HSC-44PE (upper and lower left), the fibroblasts cocultured with 44As3 versus mono-cultured fibroblasts (upper and lower middle), and the fibroblasts cocultured with HSC-44PE versus mono-cultured (upper and lower right). NES: normalized enrichment score. The *p*-value was calculated by GSEA. (**C**) A heat map shows glycolysis-related gene expression in each culture condition. n = 2 technical replicates. (**D**) qRT-PCR analysis of ENO1, ENO2, LDHA, PDK1, and PDK3 expression in iNF-58 cells cocultured with DGC cells with high metastatic potential compared to the expression in mono-cultured iNF-58 cells. n = 3 biological replicates. Error bars represent s.d. *, $p < 0.05$ from ANOVA followed by Tukey's HSD post hoc comparisons. (**E**) Western blot analysis of glycolysis-related proteins, ENO1, ENO2, LDHA, PDK1, PDK3, and β-actin in iNF58 cells in mono-culture or coculture with DGC cells (left). Densitometric analysis of Western blot on ENO2, LDHA and PDK3 normalized to the level of β-actin (right). n = 3 biological replicates. Error bars represent s.d. *, $p < 0.05$, **, $p < 0.01$ from ANOVA followed by Tukey's HSD post hoc comparisons.

with those that were in mono-culture (Fig 1B, upper right). As shown in the heat map, 44As3 significantly enhanced glycolysis-related gene expression in the fibroblast lines (Fig 1C). In contrast, GSEA also showed that "oxidative phosphorylation" pathway was significantly enriched in fibroblasts cocultured with HSC-44PE cells (Fig 1B, lower left and middle). We also run GSEA and found that the "glycolysis" pathway was significantly enriched in 44As3 compared to HSC-44PE (S1 Fig). To focus on the phenotypic changes in the fibroblasts educated by DGC cells, we performed further experiments for the fibroblasts that were cocultured with DGC cells.

The expression changes in glycolysis-related genes that were identified in the microarray data were validated by qRT-PCR and immunoblot analysis (Fig 1D and 1E). Of these genes, LDHA and ENO2 expression were remarkably increased in iNF-58 cells cocultured with

44As3 cells (Fig 1D and 1E). The LDHA expression was also significantly increased in iNF60 cocultured with 44As3 cells (S2 Fig). These data suggest that GC cells with high metastatic potential can strongly induce aerobic glycolysis in stomach fibroblasts.

## DGC cells with high metastatic potential enhanced glucose consumption and lactate production in stromal fibroblasts

To further characterize the fibroblasts cocultured with 44As3, we measured lactate production and glucose consumption of fibroblasts grown in mono-culture or coculture. Lactate

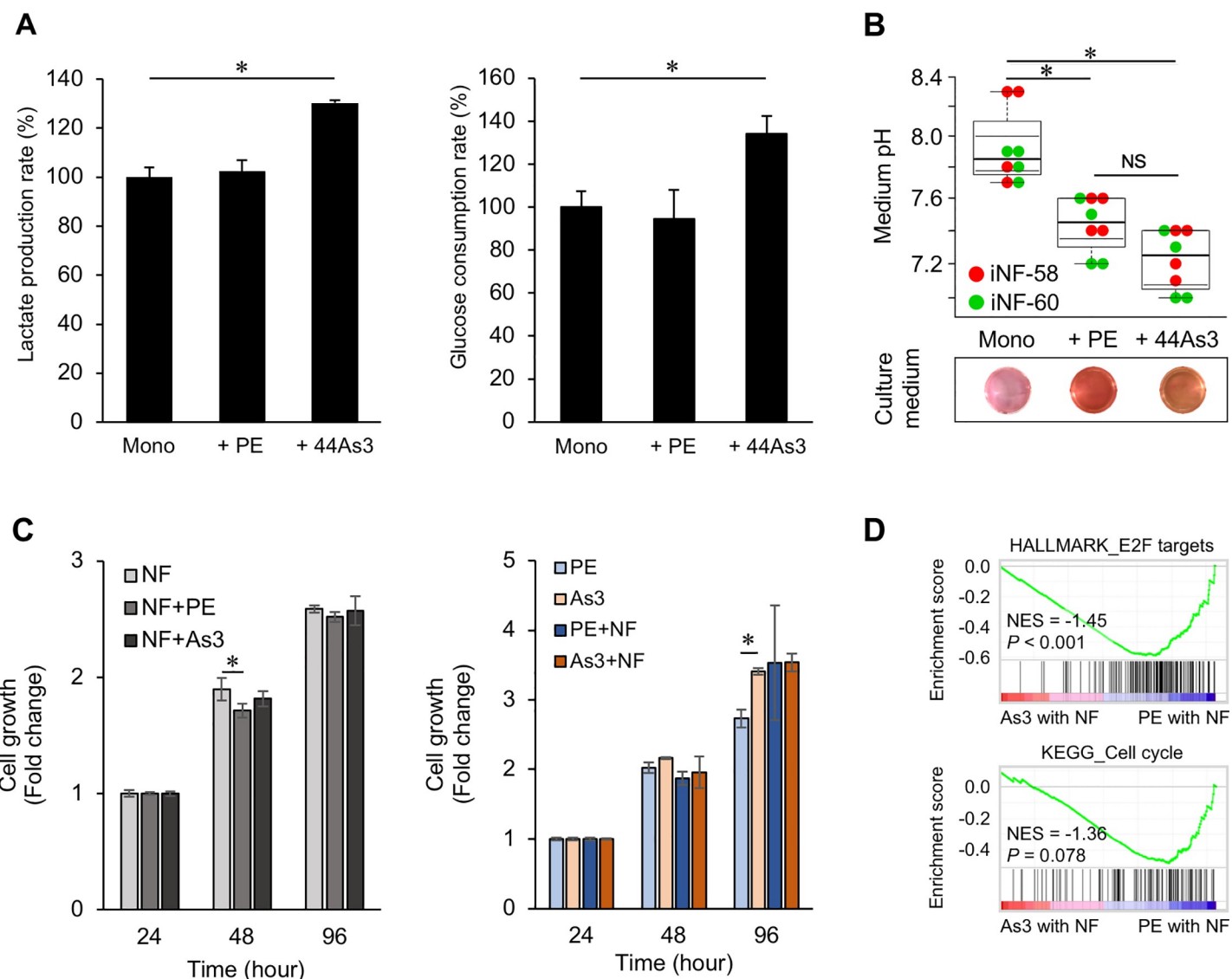

**Fig 2. DGC cells with high metastatic potential enhanced the metabolic switch to aerobic glycolysis in the fibroblasts.** (A) Quantification of lactate production and glucose consumption in cocultured or mono-cultured iNF-58 cells. n = 3 biological replicates. Error bars represent s.d. *, $p < 0.05$ from ANOVA followed by Tukey's HSD post hoc comparisons. (B) The pH of medium in which cocultured or mono-cultured iNF-58 cells and iNF-60 cells were maintained. n = 4 technical replicates in each fibroblast. Error bars represent s.d. *, $p < 0.05$ from ANOVA followed by Tukey's HSD post hoc comparisons. (C) The cell proliferation rate of iNF-58 cells (left) and DGC cell lines (right) in the mono-culture and coculture. n = 3 technical replicates. Error bars represent s.d. *, $p < 0.05$ from ANOVA followed by Tukey's HSD post hoc comparisons. (D) GSEA of 44As3 cells cultured with fibroblasts (As3 with NF) versus HSC-44PE cells cultured with fibroblasts (PE with NF), highlighting cell proliferation-related phenotypes. NES: a normalized enrichment score. The p-value was calculated by GSEA.

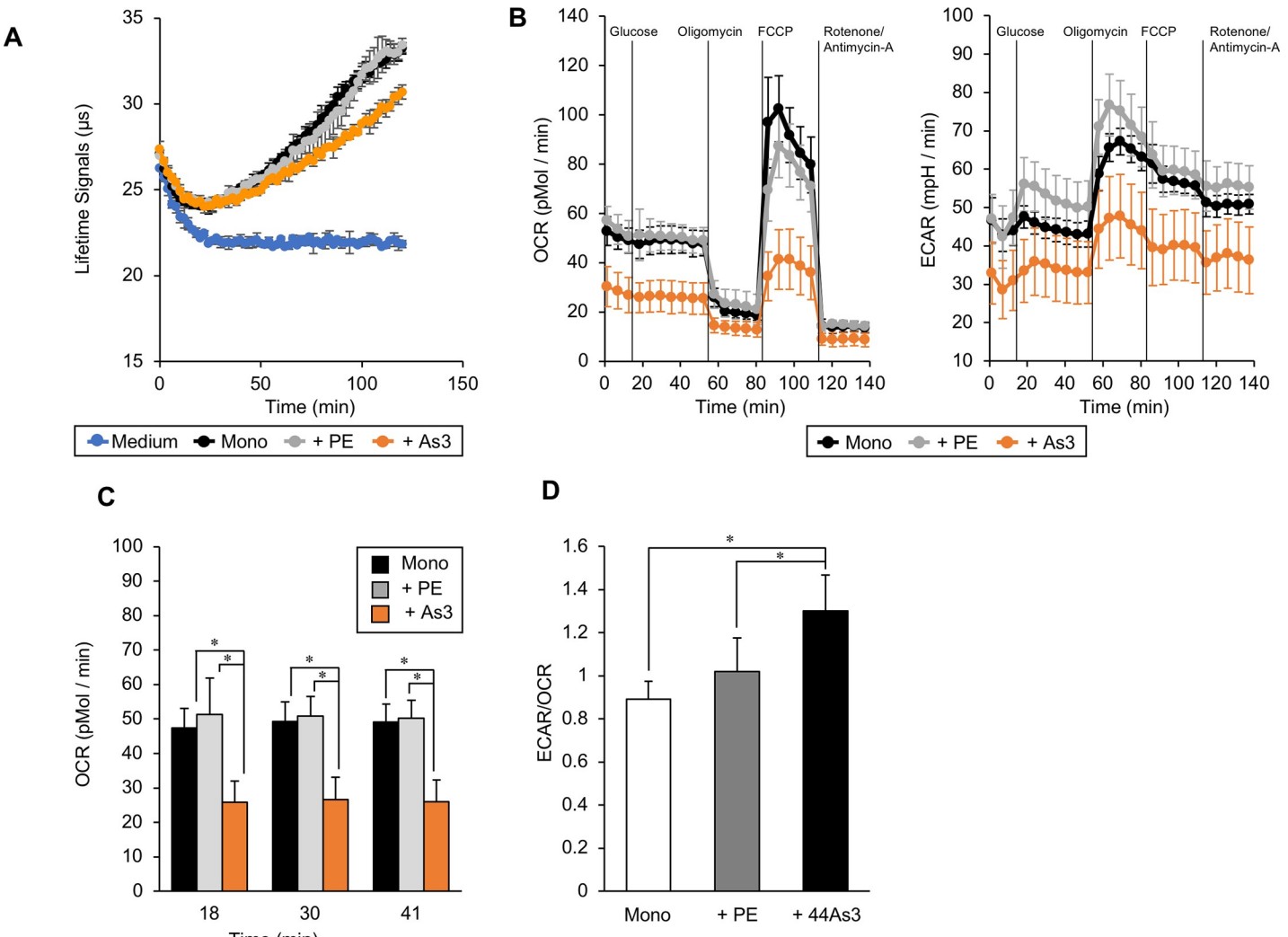

**Fig 3. DGC cells with high metastatic potential promoted the metabolic switch from oxidative phosphorylation to aerobic glycolysis in fibroblasts. (A)** The lifetimes of oxygen consumption of iNF-58 cells in mono-culture and coculture is indicated by changes in fluorescence. n = 3 technical replicates. **(B)** OCR and ECAR in iNF-58 cells cocultured with DGC cells with high metastatic potential and mono-cultured iNF-58 cells. n = 8 technical replicates. **(C)** The histogram shows the basal oxygen consumption level in iNF-58 cells at three different time points (18, 30, and 41 min). **(D)** The histogram shows the ratio of ECAR:OCR at baseline (41 min).

production and glucose consumption were increased in iNF-58 cells cocultured with 44As3 cells compared to iNF-58 cell mono-culture and cocultured with HSC-44PE cells (Fig 2A). The color of conditioned medium derived from iNF-58 cells and iNF60 cells in coculture with DGC cells turned from pink to orange, and the pH decreased (approximately 7.9 to 7.4 and to 7.2, Fig 2B). These data suggest that 44As3 cells affect glucose metabolism in fibroblasts. To exclude the possibility that a difference in the cell proliferation rate influenced the glucose metabolism of fibroblasts, we also analyzed the proliferation rate of cancer cells and fibroblasts in coculture. As shown in Fig 2C, the coculture with DGC cells did not promote cell growth in the fibroblasts (Fig 2C). While the proliferation rate of 44As3 was higher than HSC-44PE in mono-culture, there is no significant difference between HSC-44PE grown with fibroblasts and 44As3 grown with fibroblasts (Fig 2C). Given transcriptome analysis showing that "E2F targets" and "cell cycle" pathways were enriched in HSC-44PE cells grown with fibroblasts compared to 44As3 cells grown with fibroblasts (Fig 2D), HSC-44PE might be promoted their

cell growth by culturing with fibroblasts. Taken together, these results suggest that there is no relationship between cell growth and glycolysis induction by 44As3 cells in the coculture systems.

### Glucose metabolism was switched from oxidative phosphorylation to aerobic glycolysis in the fibroblasts cultured with DGC cells with high metastatic potential

To investigate the effect of 44As3 cells on mitochondrial respiration in fibroblasts, we measured the OCR of iNF-58 cells in mono-culture and in coculture with DGC cells using a MitoXpress Xtra Oxygen Consumption Assay. As shown in Fig 3A, 44As3 cells promoted a decrease in the lifetime signals, which reflects mitochondrial oxygen consumption, in iNF-58 cells compared to what was measured from HSC-44PE cells. We also determined the metabolic profile of iNF-58 cells cocultured with 44As3 cells using XF96. The activity of oxidative phosphorylation in iNF-58 cells, which is reflected by the maximum respiration capacity, also decreased when they were cocultured with 44As3 cells (Fig 3B and 3C). These observations are consistent with a previous report that basal oxygen consumption and oxidative phosphorylation decreased in CAFs following treatment with growth factors [17]. The ECAR/OCR ratio showed that 44As3 cells promoted glycolysis in iNF-58 cells (Fig 3D). These data suggest that DGC cells with high metastatic potential promote the metabolic switch to aerobic glycolysis in fibroblasts.

### DGC cell-derived EVs were not involved in the metabolic status of the fibroblasts

HIF-1α is a major transcription factor that regulates glycolysis in the tumor microenvironment [1,18,19]. Indeed, our GSEA data of the fibroblasts cocultured with 44As3 cells also showed that genes in the "Hypoxia" category were significantly changed compared with mono-cultured cells and cells cocultured with HSC-44PE cells (S3A Fig). However, at least in our DGC cell line models, cancer cell lines did not affect the stability of HIF-1α and HIF-2α in the fibroblasts (S3B Fig), suggesting that other mechanisms direct glycolytic reprogramming of the fibroblasts cocultured with 44As3 cells. As recent studies have demonstrated that EVs are novel mediators of metabolic reprogramming in CAFs [9–11], we hypothesized that cancer cell-derived EVs enhance glycolysis in fibroblasts. To this end, we used the transcriptome profile data that we previously generated [15]. However, GSEA and heat maps showed that 44As3-derived EVs did not drive the metabolic transformation of fibroblasts (S3C and S3D Fig). Therefore, these data suggest that metabolic reprogramming in fibroblasts cocultured with 44As3 cells was not induced by HIF stabilization or cancer-derived EVs.

### Discussion

Growing evidence has demonstrated that CAFs undergo aerobic glycolysis and recondition the metabolic environment to create the metastatic niche [20]. This metabolic reprogramming of CAFs can provide pivotal signals that are critical for cancer-stromal interplay and create an appropriate tumor microenvironment for cancer progression [8,17,21]. We and others have shown that CAF phenotypes vary according to cancer phenotypes and aggressiveness. For instance, four different subtypes of breast cancer cells induced four distinct CAF subtypes with distinct phenotypes and functions [13]. Breast cancer cell-derived PDGF-CC directed CAF phenotypes, which can affect cancer subtypes [22]. Mesenchymal and nonmesenchymal high grade serous ovarian cancers accumulated different CAF subtypes depending on oxidative

stress-induced miR-200 expression [14]. Our previous study demonstrated that highly metastatic cancer cell-derived EVs selectively induced different phenotypes in stomach fibroblasts [15]. However, it remains unclear whether the metastatic properties of GC cells affect the metabolic reprogramming of stromal cells. In this study, we analyzed the impact of DGC cells with high or low metastatic potential on glucose metabolism in stomach fibroblasts. DGC cells with high metastatic potential significantly changed glycolysis-related gene expression in the fibroblasts. Consistent with these data, we also showed that lactate production, glucose consumption and the OCR were altered in fibroblasts cultured with cancer cells with high metastatic potential. Interestingly, cancer cells with low metastatic potential could not induce such metabolic reprogramming in the fibroblasts, suggesting that the metabolic switch to glycolysis in stromal fibroblasts may reflect the metastatic properties of GC cells. Additional studies using clinical samples should be performed to clarify the potential relevance of metabolic status in CAFs to patient prognosis.

Although our study elucidated important properties of DGC cells with high metastatic potential on glucose metabolism in fibroblasts, the mechanisms responsible for switching from oxidative phosphorylation to aerobic glycolysis in fibroblasts are unknown. Several factors have been reported as regulators of glucose metabolism in the tumor stroma. miRNAs and lncRNAs can be delivered by EVs to alter the metabolic environment of the metastatic niche [9–11]. TGF-β and PDGF induce aerobic glycolysis in CAFs through IDH3α inhibition and HIF-1α stabilization [17]. However, at least in our model, metabolic reprogramming was induced without changes in HIF-1α and HIF-2α stabilization or in the involvement of cancer-derived EVs, suggesting that other humoral factors are associated with the glucose metabolism of fibroblasts. We previously showed that TNF-α signaling-related gene expression was changed in fibroblasts cultured with cancer cells with high metastatic potential [15]. Since several reports have shown that TNF-α likely regulates glucose metabolism in cancer cells without HIF-1α stabilization [23,24], the TNF-α signaling pathway might be involved in the induction of aerobic glycolysis in fibroblasts. Further examination should be performed to completely clarify the molecular mechanisms of metabolic reprogramming in fibroblasts by cancer cells with high metastatic potential.

In conclusion, we demonstrated the differential abilities of GC cells with high and low metastatic potential to metabolically reprogram fibroblasts. We believe that our findings will lead to an understanding of the molecular basis of CAFs and the improvement of cancer diagnosis.

## Supporting information

**S1 Fig. Gastric cancer cells with high metastatic potential increased the expression level of glycolysis-related genes in the cocultured iNF-60 cells.** Western blot analysis of glycolysis-related proteins, ENO1, ENO2, LDHA, PDK1, PDK3, and β-actin in the cells cocultured with DGC cells with high metastatic potential or in the mono-cultured iNF60 cells (left). Densitometric analysis of Western blot on ENO2, LDHA and PDK3 normalized to the level of ACTB (right). Error bars represent s.d. *, $p < 0.05$ from ANOVA followed by Tukey's HSD post hoc comparisons.
(TIFF)

**S2 Fig. GSEA of the 44As3 versus HSC-44PE that were in mono-cultured.** NES: normalized enrichment score. The $p$-value was calculated by GSEA.
(TIFF)

**S3 Fig. Cancer-derived extracellular vesicles were not able to induce a metabolic shift in the fibroblasts. (A)** GSEA of the fibroblasts cocultured with 44As3 cells versus mono-cultured

fibroblasts (left) and the fibroblasts cocultured with 44As3 cells versus those cocultured with PE (right), highlighting a hypoxia-related phenotype. NES: a normalized enrichment score. The p-value was calculated by GSEA. **(B)** Western blot analysis of HIF-1α and HIF-2α in fibroblasts cocultured with DGC cells with high metastatic potential or mono-cultured iNF-58 cells. **(C)** GSEA of the fibroblasts treated with 44As3-derived extracellular vesicles (EVs) (44As3 EVs) versus those treated with HSC-44PE (PE EVs) (upper) and the fibroblasts treated with 44As3-derived extracellular vesicles (44As3 EVs) versus PBS-treated fibroblasts (PBS) (lower), highlighting glucose metabolism and oxidative phosphorylation phenotypes. NES: a normalized enrichment score. The p-value was calculated by GSEA. **(D)** Heat map showing glycolysis-related and oxidative phosphorylation-related gene expression in each condition. n = 2 technical replicates.
(TIFF)

**S4 Fig. Uncropped full-length pictures of western blotting membranes in Fig 1E, S2 and S3 Figs.** Membranes were often cut to enable blotting for multiple antibodies. The blue dotted squares from raw blots were shown in the main text.
(PDF)

## Acknowledgments

We thank Dr. Tai Kudo in PrimeTech Co., Ltd. for his excellent technical assistance.

## Author Contributions

**Conceptualization:** Yusuke Yamamoto.

**Data curation:** Akiko Kogure, Yutaka Naito, Yusuke Yamamoto.

**Formal analysis:** Akiko Kogure, Yutaka Naito, Yusuke Yamamoto.

**Funding acquisition:** Yutaka Naito, Yusuke Yamamoto, Takahiro Ochiya.

**Investigation:** Akiko Kogure, Yutaka Naito.

**Methodology:** Masakazu Yashiro, Tohru Kiyono, Kazuyoshi Yanagihara, Kosei Hirakawa.

**Project administration:** Takahiro Ochiya.

**Resources:** Masakazu Yashiro, Tohru Kiyono, Kazuyoshi Yanagihara, Kosei Hirakawa.

**Writing – original draft:** Akiko Kogure, Yutaka Naito, Yusuke Yamamoto.

**Writing – review & editing:** Akiko Kogure, Yutaka Naito, Yusuke Yamamoto, Masakazu Yashiro, Tohru Kiyono, Kazuyoshi Yanagihara, Kosei Hirakawa, Takahiro Ochiya.

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
