## [Decision Letter · Decision Letter 0]

20 Nov 2019

PONE-D-19-29915

Cancer cells with high-metastatic potential promote a glycolytic shift in activated fibroblasts

PLOS ONE

Dear Dr. Ochiya,

Thank you for submitting your manuscript to PLOS ONE. After careful consideration, we feel that it has merit but does not fully meet PLOS ONE’s publication criteria as it currently stands. Therefore, we invite you to submit a revised version of the manuscript that addresses the points raised during the review process. Mainly, the manuscript is not technically sound at this point and it can only be reconsidered if you include appropriate experiments performed in triplicates as suggested by the reviewers.

We would appreciate receiving your revised manuscript by 1st May 2020. To enhance the reproducibility of your results, we recommend that if applicable you deposit your laboratory protocols in protocols.io, where a protocol can be assigned its own identifier (DOI) such that it can be cited independently in the future. For instructions see: http://journals.plos.org/plosone/s/submission-guidelines#loc-laboratory-protocols

We look forward to receiving your revised manuscript.

Kind regards,

Pankaj K Singh, Ph.D.

Academic Editor

PLOS ONE

Journal Requirements:

2. Please provide additional information about each of the cell lines used in this work, including the source and any quality control testing procedures (authentication, characterisation, and mycoplasma testing). For more information, please see http://journals.plos.org/plosone/s/submission-guidelines#loc-cell-lines.

3. To comply with PLOS ONE submission guidelines, in your Methods section, please provide additional information regarding your statistical analyses. For more information on PLOS ONE's expectations for statistical reporting, please see https://journals.plos.org/plosone/s/submission-guidelines.#loc-statistical-reporting.

Reviewers' comments:

Reviewer's Responses to Questions

**Comments to the Author**

1. Is the manuscript technically sound, and do the data support the conclusions?

Reviewer #1: No

Reviewer #2: Partly

2. Has the statistical analysis been performed appropriately and rigorously? 

Reviewer #1: No

Reviewer #2: No

3. Have the authors made all data underlying the findings in their manuscript fully available?

Reviewer #1: Yes

Reviewer #2: Yes

4. Is the manuscript presented in an intelligible fashion and written in standard English?

Reviewer #1: No

Reviewer #2: Yes

5. Review Comments to the Author

Reviewer #1: In the manuscript entitled “Cancer cell with high-metastatic potential promote a glycolytic shift in activated fibroblasts, Kogure et. al. investigate the metabolic interaction between highly metastatic gastric cancer cells and CAFs. Previous studies have shown that CAFs exhibit elevated glycolytic metabolism, which may contribute to cancer cell survival and proliferation within metabolically challenging microenvironment. This study made an interesting observation that gastric cancer cells with high metastatic capacity induced glycolytic gene expression in cocultured fibroblasts. The authors further attempted to elucidate potential mechanisms underlying cancer cell-derived glycolytic reprogramming of CAFs (e.g. cancer cell-derived EVs and hypoxia/HIFs).

While this is an interesting observation, as current study stands the experimental data do not necessarily support what the authors are proposing. It is unclear how the coculture system employed can be properly interpreted. For instance, in Figure 2, lactate in the co-culture system is supplied by both cancer cells and fibroblasts. Accordingly, are HSC-44PE cells more glycolytic than 44As3? It is very critical to characterize and present metabolic features (e.g. glycolysis) of cancer cells as well.

It is also very confusing how the authors performed the Kaplan-Meier survival analysis. It seems that the LDHA and ENO2 expression is not specific to CAFs. If so, how this survival data can support the hypothesis that the authors are proposing?

The previous study published by the same group was cited twice (#15 and 16).

Reviewer #2: In the manuscript, the authors demonstrate differential metabolic reprogramming in fibroblast metabolism upon co-culture with cancer cells displaying varying degree of metastatic potential. Cancer cells with high metastatic potential upregulate glycolysis in fibroblasts and decrease oxidative phosphorylation. Although the finding is interesting, further experiments are required to validate the results. The authors need to address following concerns before publication:

1. Most of the experiments are done with two replicates, the experiments need to be performed with at least three technical replicates.

2. In Fig 1B, the GSEA comparisons are made between As3 with PE and As3 with monoculture. The GSEA analysis between PE and monocultures need to be included. What is the status of enrichment in glycolysis and oxidative phosphorylation in PE versus monoculture.

3. In Fig1E, the densitometric analysis of blots needs to be added to highlight the differences. The authors should repeat the co-culture experiment with other fibroblast cells (iNF-58).

4. In Fig2C, the cell growths should be compared with the monocultures of fibroblasts and cancer cells. Also, the statistics is missing in the figure.

5. The authors have correlated levels of genes Ldha and Eno2 with survival of GC patients. However, it will be pertinent to show the expression of Ldha and Eno1 in fibroblast in stromal cells of cancer patient's specimens. Also, it will be interesting to note if there is any survival difference between the patients classified on the basis of glycolytic gene expression in stromal cells.

6. PLOS authors have the option to publish the peer review history of their article (what does this mean?). If published, this will include your full peer review and any attached files.

Reviewer #1: No

Reviewer #2: No

---

## [Author Response · Author response to Decision Letter 0]

11 May 2020

Point-by-point response to each of the reviewer’s comments

We are grateful to all of the reviewers for their critical comments and insightful suggestions that have helped us considerably improve our paper. As indicated in the responses below, we have taken all of these comments and suggestions into account in the revised version of our paper, including the supplementary information.

Response to the comments from Reviewer #1

Q1. It is unclear how the coculture system employed can be properly interpreted. In Figure 2, lactate in the co-culture system is supplied by both cancer cells and fibroblasts. 

A1. We apologize that we didn’t properly describe it in the text. We applied the transwell culture system for the coculture of the fibroblasts and DGC cells. As the reviewer#1 pointed out, both DGC cells and fibroblasts enabled to supply the lactate with each other during incubation. However, for lactate production assay, we firstly removed the chamber including cancer cells, and washed them with PBS. Then, we replaced fresh media with 1g/L glucose and incubated these fibroblasts cocultured with DGC cells. We measured the lactate level in the cultured media of fibroblasts in 3 hr after the medium change. On these methods, we believe that there is no lactate derived from DGC cells left over. We added these methods in the Materials and Methods section of “Analysis of lactate production, glucose consumption and pH.”

Q2. Are HSC-44PE cells more glycolytic than 44As3? It is very critical to characterize and present metabolic features (e.g. glycolysis) of cancer cells as well.

A2. This is an important comment. To address this point, we also run GSEA and found that the "glycolysis" pathway was significantly enriched in 44As3 compared to HSC-44PE (Figure R1). Therefore, it suggests that similar glucose metabolism features of highly metastatic DGC cells were induced in the fibroblasts by coculture. We added these important findings in the Fig S1 and main text (Page 8, line 20 to 23). However, since we would like to focus on the metabolic features on the fibroblasts surrounding cancer cells, we have not performed additional experiments to characterize cancer cells in this manuscript.

Q3. It is also very confusing how the authors performed the Kaplan-Meier survival analysis. It seems that the LDHA and ENO2 expression is not specific to CAFs. If so, how this survival data can support the hypothesis that the authors are proposing?

A3. We completely agree with reviewer's comment because Kaplan-Meier plotter data are based on gene expression profiles from the bulk tumour tissues. The reviewer #2 also gave us a similar comment. These data do not directly reflect the relationship between patient outcome and expression of LDHA and ENO2 in the fibroblasts within tumour microenvironment. Therefore, we removed Fig.4 from the revised manuscript to avoid misunderstanding for the readers.

Q4. The previous study published by the same group was cited twice (#15 and 16).

A4. We do appreciate your comment. We corrected the references.

Response to the comments from Reviewer #2

Q1. Most of the experiments are done with two replicates, the experiments need to be performed with at least three technical replicates.

A1. This is an important comment. We performed all experiments with at least three technical replicates and we added them in the figures (Fig1 D and E, Fig2 B). Following these changes, we also edited the text in the result section (Page 9, line 24 to 27)

Q2. In Fig 1B, the GSEA comparisons are made between As3 with PE and As3 with monoculture. The GSEA analysis between PE and monocultures need to be included. What is the status of enrichment in glycolysis and oxidative phosphorylation in PE versus monoculture.

A2. Thank you very much for your valuable comment. We also performed the GSEA analysis between fibroblasts cocultured with HSC-44PE and those in mono-culture. As shown in Figure R2, GSEA showed that significant enrichment of genes with a role in “glycolysis” functions in fibroblasts cocultured with HSC-44PE cells compared to fibroblasts that were in mono-culture. On the other hand, the fibroblasts cocultured with HSC-44PE showed the tendency to associated with gene enrichment of "oxidative phosphorylation" although there are no significant differences. These results suggest that HSC-44PE also affects glucose metabolism in the fibroblasts. However, "glycolysis" functions were significantly enriched in the fibroblasts cocultured with 44As3 compared with that with HSC44PE, suggesting that different glucose metabolism would be induced in the fibroblasts by high-metastatic and low-metastatic GC cells. We added these data in Fig.1B and edited the text in the result part (Page 8, line 14 to 16)

Q3. In Fig1E, the densitometric analysis of blots needs to be added to highlight the differences. The authors should repeat the co-culture experiment with other fibroblast cells (iNF-58). 

A3. We do appreciate your comment. We performed western blot by using iNF-58 and iNF-60. Then, ENO2, LDHA, and PDK3 expressions were evaluated by densitometric analysis. As shown in Figure R3, ENO2 and LDHA expressions were significantly increased in the iNF-58 cocultured with 44As3 cells. On the other hands, in iNF-60 with 44As3, only LDHA expression was increased. This observation might be due to the derived from different patient of iNF-58. These data were added in the revised Fig.1E and Fig.S2, and main text (Page 8, line 27 to 28).

Q4. In Fig2C, the cell growths should be compared with the monocultures of fibroblasts and cancer cells. Also, the statistics is missing in the figure.

A4. Thank you very much for your valuable comment. We also checked the proliferation rate in both DGC cells and fibroblasts that were in mono-culture and compared with those in co-culture. As shown in Figure R4, the coculture did not promote cell growth in the fibroblasts (Figure R4A). The proliferation rate of mono-cultured 44As3 was higher than HSC-44PE (Figure R4B). However, there is no significant difference between HSC-44PE grown with fibroblasts and 44As3 grown with fibroblasts. These data suggest that there is no relevance between glycolysis induction in the fibroblasts by DGC cells and cell growth. We added these data in the revised Fig.2C and edited the text in the result section (Page 9, line 29 to 33, and Page 10 line 2 to line 5.)

Q5. The authors have correlated levels of genes Ldha and Eno2 with survival of GC patients. However, it will be pertinent to show the expression of Ldha and Eno1 in fibroblast in stromal cells of cancer patient's specimens. Also, it will be interesting to note if there is any survival difference between the patients classified on the basis of glycolytic gene expression in stromal cells.

A5. According to the reviewer #1’s comment, Kaplan-Meier plotter data are based on gene expression profiles from the bulk tumour tissues. These data do not reflect the relationship between patient outcome and expression of LDHA and ENO2 in the fibroblasts. Since GC tissue samples were limited, it was difficult to use the samples for an immunohistochemistry analysis. Therefore, we removed Fig.4 from the revised manuscript.

---

## [Decision Letter · Decision Letter 1]

1 Jun 2020

Cancer cells with high-metastatic potential promote a glycolytic shift in activated fibroblasts

PONE-D-19-29915R1

Dear Dr. Ochiya,

We are pleased to inform you that your manuscript has been judged scientifically suitable for publication and will be formally accepted for publication once it complies with all outstanding technical requirements.

With kind regards,

Pankaj K Singh, Ph.D.

Academic Editor

PLOS ONE

Additional Editor Comments (optional):

Reviewers' comments:

Reviewer's Responses to Questions

**Comments to the Author**

1. If the authors have adequately addressed your comments raised in a previous round of review and you feel that this manuscript is now acceptable for publication, you may indicate that here to bypass the “Comments to the Author” section, enter your conflict of interest statement in the “Confidential to Editor” section, and submit your "Accept" recommendation.

Reviewer #1: All comments have been addressed

Reviewer #2: All comments have been addressed

2. Is the manuscript technically sound, and do the data support the conclusions?

Reviewer #1: Yes

Reviewer #2: Yes

3. Has the statistical analysis been performed appropriately and rigorously? 

Reviewer #1: Yes

Reviewer #2: Yes

4. Have the authors made all data underlying the findings in their manuscript fully available?

Reviewer #1: Yes

Reviewer #2: Yes

5. Is the manuscript presented in an intelligible fashion and written in standard English?

Reviewer #1: Yes

Reviewer #2: Yes

6. Review Comments to the Author

Reviewer #1: All major concerns and comments raised by the reviewer #1 have been appropriately addressed and the manuscript has been revised accordingly.

Reviewer #2: The authors have addressed most of the raised concerns satisfactorily and the manuscript can be accepted in the modified form.

7. PLOS authors have the option to publish the peer review history of their article (what does this mean?). If published, this will include your full peer review and any attached files.

Reviewer #1: No

Reviewer #2: No

---

## [Editor Report · Acceptance letter]

8 Jun 2020

PONE-D-19-29915R1 

Cancer cells with high-metastatic potential promote a glycolytic shift in activated fibroblasts 

Dear Dr. Ochiya:

I'm pleased to inform you that your manuscript has been deemed suitable for publication in PLOS ONE. Congratulations! Your manuscript is now with our production department. 

Kind regards, 

on behalf of

Dr. Pankaj K Singh 

Academic Editor

PLOS ONE